# Enhancing the Cell-Free Expression of Native Membrane Proteins by In Silico Optimization of the Coding Sequence—An Experimental Study of the Human Voltage-Dependent Anion Channel

**DOI:** 10.3390/membranes11100741

**Published:** 2021-09-28

**Authors:** Sonja Zayni, Samar Damiati, Susana Moreno-Flores, Fabian Amman, Ivo Hofacker, David Jin, Eva-Kathrin Ehmoser

**Affiliations:** 1Department of Nanobiotechnology, Institute for Synthetic Bioarchitectures, University of Natural Resources and Life Sciences, Vienna (BOKU), Muthgasse 11, A-1190 Wien, Austria; sonja.zayni@boku.ac.at; 2Department of Biochemistry, Faculty of Science, King Abdulaziz University, Jeddah 21413, Saudi Arabia; sdamiati@kau.edu.sa or; 3Science for Life Laboratory, Department of Protein Science, Division of Nanobiotechnology, KTH Royal Institute of Technology, 171 21 Stockholm, Sweden; 4Independent Researcher, A-1190 Vienna, Austria; smf8097@gmail.com; 5Department of Theoretical Chemistry, University of Vienna, Währinger Straße 17, A-1090 Wien, Austria; ivo@tbi.univie.ac.at; 6Research Group Bioinformatics and Computational Biology, Faculty of Computer Science, University of Vienna, Währinger Straße 29, A-1090 Wien, Austria; 7Avalon Globocare Corp., 4400 Route 9 South, Suite 3100, Freehold, NJ 07728, USA; david@avalon-globocare.com

**Keywords:** cell-free membrane protein expression, translation enhancer, translation initiation, ribosome docking site, sequence design

## Abstract

Membrane proteins are involved in many aspects of cellular biology; for example, they regulate how cells interact with their environment, so such proteins are important drug targets. The rapid advancement in the field of immune effector cell therapy has been expanding the horizons of synthetic membrane receptors in the areas of cell-based immunotherapy and cellular medicine. However, the investigation of membrane proteins, which are key constituents of cells, is hampered by the difficulty and complexity of their in vitro synthesis, which is of unpredictable yield. Cell-free synthesis is herein employed to unravel the impact of the expression construct on gene transcription and translation, without the complex regulatory mechanisms of cellular systems. Through the systematic design of plasmids in the immediacy of the start of the target gene, it was possible to identify translation initiation and the conformation of mRNA as the main factors governing the cell-free expression efficiency of the human voltage-dependent anion channel (VDAC), which is a relevant membrane protein in drug-based therapy. A simple translation initiation model was developed to quantitatively assess the expression potential for the designed constructs. A scoring function that quantifies the feasibility of the formation of the translation initiation complex through the ribosome–mRNA hybridization energy and the accessibility of the mRNA segment binding to the ribosome is proposed. The scoring function enables one to optimize plasmid sequences and semi-quantitatively predict protein expression efficiencies. This scoring function is publicly available as webservice XenoExpressO at University of Vienna, Austria.

## 1. Introduction

Understanding the structure and function of membrane proteins is key in many biological processes yet faces numerous issues. Membrane proteins are notoriously difficult to synthesize: in cells, membrane proteins are usually expressed in low amounts, and their expression profile is heavily controlled as part of regulatory processes. In addition, in-cell expression of recombinant membrane proteins only works for those proteins that do not significantly alter the physiology of their hosts. The characterization of membrane proteins is no less difficult: the structural integrity of membrane proteins is hard to preserve in extracellular conditions, and function may be lost if proteins are removed from their native membranes.

The production of membrane proteins outside living cells circumvents many of the issues of in-cell synthesis [1,2]. Cell-free synthesis uses cell lysates to generate in situ rightly folded membrane proteins [3] from exogeneous mRNA or DNA, which can be directly incorporated into artificial membranes [4,5,6,7]. The functionalization of nanodiscs with VDAC receptor via cell-free protein synthesis was shown recently and in very detailed analysis [8], also supporting the in-principle functionality of cell-free in vitro synthesized membrane proteins.

However, cell-free and in-cell synthesis face a common challenge. The design of the plasmid vector is crucial. This genetic construct lodges the sequences of the transcription promotor, of the ribosomal binding site, RBS, and occasionally of translation enhancers in addition to the target gene [1,9]. The sequence layout, particularly in the vicinity of the gene’s initiation or start codon, has become the quintessence of cell-free protein expression, yet it has not been fully exploited in optimizing constructs for protein expression. The coding region adjacent to the start codon remains untapped in both the in silico [10,11] and wet-bench design of constructs, and finding a working construct is to date mainly based on trial and error.

Herein, we present a rationalized approach to the generation of constructs for the expression of wild-type, human membrane proteins in prokaryotic cell-free systems, which includes alterations in the coding sequence proximal to the start codon. Our strategy possibly paves the way for efforts in the hydrophilization of membrane proteins on the protein level, using codon engineering as a powerful strategy to achieve functionally folded membrane proteins [5]. As a relevant example for improving expression levels, we chose the human voltage-dependent anion channel or VDAC, a small, 285 amino acid-long protein (M_w_ = 31 kDa) that is predominantly found in the mitochondrial outer membrane [12,13]. VDAC forms cylindrical channels across the membrane with a 20–30 Å diameter, allowing the passage of ions and small molecules [13,14,15], and it is involved in various pathophysiological mechanisms. 

The functionality of our example of choice, the in vitro synthesized VDAC receptor, has been demonstrated by functional characterization assays in planar membrane architectures [5]. In our previously published work, we assessed the functional ion channel activity of VDAC protein in real time via the combination of quartz crystal microbalance with dissipation monitoring (QCM-D) and electrochemical impedance spectroscopy (EIS), where changes in the electrical resistance and capacitance of S-layer supported lipid membranes indicated the functional reconstitution of VDAC-I-A protein into synthetic membranes [6], which recently was also shown by Nibali and DePinto et al. [8]. For functional VDAC protein expression, we thought about a plausible and straightforward cloning strategy, employing the commercial Gateway^®^ recombination technology. This comprises a two-step reaction: the first step is the recombination of a PCR product into a generic Donor Vector (pDONR221), and in the second step, expression vectors with custom-designed special features, such as a host, resistances, efficiencies, tags, etc., can be chosen. We identified pDEST14, a standard expression vector, which is suitable for N-terminal tag-free expression for our protein of interest. We involved pDEST17, coding for an N-terminally His-tagged VDAC protein, where the N-terminal His-tag was probed as an expression enhancer. In sum, we present the in silico-optimized expression of the VDAC membrane protein in pDEST14 to demonstrate the potential of cell-free protein expression enhancement through rationally designed synonymous mutations of the coding sequence proximal to the start codon, leaving the translated protein product N-terminally untagged, e.g., unmodified, and thus enabling work with wild-type membrane proteins.

## 2. Materials and Methods

Cloning and purification of plasmids. Cloning was performed with Gateway^®^ recombination cloning technology (Invitrogen, Thermo Fisher Scientific, Waltham, MA, USA). Eight forward primers and one reverse primer were designed [16]. The DNA of VDAC (855 base pairs) was amplified by PCR (Biometra Thermocycler, Analytik Jena, Jena, Germany), with Phusion DNA polymerase (Thermo Fisher) and vector pQE30-VDAC as a template. All PCR products were purified with the MinElute PCR purification kit (Qiagen, Venlo, The Netherlands). Gateway^®^ recombination was performed with enzyme mixes BP Clonase II and LR Clonase II according to the manufacturer’s instructions (Invitrogen, Thermo Fisher Scientific). BP reactions were carried out with the purified fragments and the entry vector pDONR221. LR reactions were performed with entry clones from individual bacterial colonies and destination vectors pDEST14 and pDEST17. BP and LR products were subsequently transformed into *E. coli* strains DH5α and Top 10 (Invitrogen). Positive clones were identified by in situ PCR (RedTaq Master Mix, Sigma-Aldrich, St Louis, MO, USA). Then, plasmid DNA was purified with the QIAprep Spin Miniprep Kit (Qiagen) and examined through digestion with restriction enzymes *EcoRI*/*HindIII* and *PstI*/*Xhol* for DONR and DEST vector constructs, respectively (Thermo Fisher). Sequencing of VDAC gene inserts for DONR and DEST vectors was performed with the VDAC-specific and T7 promotor/T7 terminator primers (LGC Genomics, Berlin, Germany; Microsynth, Balgach, Switzerland), respectively. Plasmids were purified with Midi prep kits (Qiagen Plasmid Midi Kit or innu PREP Plasmid MIDI Direct Kit, Analytik Jena). 

Cell-free synthesis. Reactions were performed with two different kits, the S30 T7 High-yield protein expression system (Promega, Fitchburg, MA, USA) and the PURExpress^®^ in Vitro Protein Synthesis Kit (New England BioLabs, Ipswich, SD, USA) according to the manufacturer’s instructions. The results herein reported refer to those obtained with the BioLabs kit, as it proved the most effective. Two hundred and fifty nanograms of plasmid, 0.2 µL of Ribonuclease inhibitor (RNasin, Promega, Austria), and 0.4 µL of FluoroTect TM Green_Lys_ were added to PURExpress extracts to a reaction volume of 10 µL. After two hours of incubation at 37 °C, 10 µL of sample dilution buffer (LDS sample buffer reducing agent, Invitrogen, Thermo Fisher) were added to the mixture. Protein denaturation in the diluted samples was conducted at 70 °C for 10 min before electrophoresis.

SDS-PAGE and Western Blot. The denatured samples were loaded into 10% precast gels (Invitrogen, Thermo Fisher). Electrophoresis was conducted at a constant potential of 200 V for 45 min and imaged immediately using the Safe Imager 2.0 TM Blue Light Transilluminator. Every expression construct has been repeated for probing its protein-synthesis efficiency. The resulting gels show examples of preparations with emphasis on relative protein amounts in regard to the standard but more importantly in relation to each other. The emission fluorescence at ≈470 nm of the fluorescent lysine accounted for the optical visualization of the protein bands. Two identical acrylamide gels were prepared: (1) for the staining procedure, Coomassie staining was performed with SimplyBlueTM SafeSTain solution (Invitrogen Thermo Fisher); (2) for Western blotting, after electromediated protein transfer from the gel to the PVDF membrane (iBlot^®^, Thermo Fisher), the immunodetection of proteins was carried out in an InfraRed Imager (Odyssey^®^ Infrared Imaging System, LI-COR Biosciences, Lincoln, NE, USA), using rabbit monoclonal anti-VDAC (Cell Signaling Technology, Cambridge, UK) or anti-6x HIS-tag (Gen Tex) as a primary antibody and goat anti-rabbit IRDye 680 (LI-COR) as a secondary antibody. PageRulerTM Plus Prestained Protein Ladder (Thermo Fisher) were used as a standard.

RNA detection and quantitative PCR (qPCR). Levels of RNA were measured with a ND-10000 Spectrophotometer (Nanodrop Technologies, Wilmington, DE, USA) on RNA-isolated samples [17]. For qPCR, ≈650 ng of isolated RNA was reverse-transcribed into cDNA with the iScript TM Select cDNA synthesis kit and random primers (Bio-Rad, Hercules, CA, USA). qPCR was performed in a 48-well MiniOpticon Real-Time PCR System (Bio-Rad) on sample triplicates (20 µL total reaction volume) [15]. An AdvancedTM Universal SYBR Green Supermix (Bio-Rad) was used to prepare the master mix for each primer.

Calculation of *δ* and in silico optimization of mRNA constructs. Hybridization and opening energies, Δ*E_SD_* and Δ*E_open_*, were calculated with *RNAduplex* and *RNAup*, respectively [17,18]. Δ*E_tRNA_* was added as a stabilizing constant, −1.19 kcal/mol or −0.075 kcal/mol, only for start codons AUG or GUG, respectively. Optimization was conducted with a self-devised simulated annealing algorithm that performs, selectively accepts, and characterizes random single-nucleotide swaps in source transcripts. Δ*E_open_* (i) for single and sets of constructs, respectively, were calculated with *RNAplfold* from genome sequences available at microbes.uscs.edu and ensemble_biomart.

## 3. Results

The design of expression constructs was performed not only to enable an assessment of the influence from expression modulators on protein expression efficiency but also to assign their optimal location upstream and downstream in relation to the initiation codon. The generation of constructs was accomplished by the recombination of a PCR product into commercially available plasmids. Consequently, the PCR product starts with a specific nucleotide sequence, which is called a primer. The target sequence in our example is represented by the VDAC-encoding gene. By the introduction of various self-designed primers, we were able to modify the genetic code in a controlled fashion and hence assess the effect of these modifications on protein expression. The original pDEST17 plasmids provide sequences before (upstream) and after (downstream) the ATG codon in the untranslated and translated regions, 5′UTR and TR, respectively (Figure 1a). The UTR is preceded by the T7 promoter and lodges a prokaryotic RBS in the form of a Shine–Dalgarno (SD) sequence. The TR starts with a 26 amino acid-long sequence containing a 6x HIS-tag (Figure 1b). pDEST17 allows the insertion of the PCR product right after this sequence in the TR. Consequently, the 5′UTR and the location of the RBS is fixed.

Figure 1c shows that pDEST17-based constructs (VDAC-I) enable protein expression, and alterations in the genetic code far downstream of the start codon have no significant effect on the protein expression efficiency. The SDS-PAGE gel and Western blot [19] of the reaction mixture, after protein expression with constructs VDAC-I-A and VDAC-I-B and in the presence of fluorescent lysine, display a single band at approximately 39 kDa of similar intensity. This is indicative of similar expression levels of a single protein. Protein characterization via MALDI-TOF mass spectrometry of trypsin-digested protein fragments reveal that 34% of the peptides match VDAC sequences, which is sufficient to confirm the primary structure of VDAC. The insertion of the chloramphenicol acetyltransferase (CAT)-enhancer sequence [20], as in VDAC-I-C, does not significantly increase the level of protein expression. The results point to the HIS-tag-containing sequence, possibly in combination with the RBS-starting sequence, as the essential cause for VDAC expression.

Our first hypothesis sets the length and nature of the untranslated region between the T7 promoter and the start codon, the 5′UTR, as decisive in gene transcription and translation, so we employed the plasmid pDEST14 to gain better control over this region, while aiming to express native, tag-free VDAC at comparable levels to those attained through pDEST17-based constructs. pDEST14 provides the T7 promoter as its pDEST17 counterpart does; however, unlike the latter, it allows the insertion of self-designed primers at desired locations upstream and downstream the start codon.

Figure 2 shows the primer sequences of the pDEST14-based constructs (VDAC-II) and their respective VDAC expression levels. Although the primer sequences of VDAC-II-A and VDAC-II-B are in turn identical to those of VDAC-I-A and VDAC-I-B, there is hardly evidence of protein expression, as shown by the SDS-page gel fluorescence scan example (Figure 2c) and the corresponding Western blot [19]. This evinces the enhancer role of the pre-VDAC sequence and the 5′UTR in pDEST17-based plasmids.

In view of these results, we directed our efforts toward investigating the role of the 5′UTR and the adjacent TR in protein expression. Starting at the SD sequence, we inserted the 5′UTR of the pDEST17 vector into pDEST14-based constructs at the same location. The resulting construct, VDAC-II-C, enables marginal protein expression, as evinced by the appearance of a weak band above 36 kDa (Figure 2c). The construct VDAC-II-D does as well, with the same first three-codon-long coding sequence of the pDEST17 vector. Only the insertion of the four-codon-long CAT-enhancer sequence after the start codon, as in VDAC-II-E and II-F, increases the levels of protein expression, irrespectively of the 5′UTR choice. However, in this case, VDAC expression is enhanced at the expense of capping the N-terminus of the protein sequence with the non-native amino acid sequence EKKI.

At this point, it is crucial to consider whether cell-free VDAC expression is hampered at the transcriptional or translational level. Should gene transcription determine protein expression, mRNA levels would be significantly higher in those cases where protein is expressed than in those where expression is marginal or not detected. In other words, any changes in levels of transcription by T7 polymerase should result in changes in the levels of protein expression. Figure 3 shows that this is not the case; quantitative PCR (Cq values) of cDNA derived from transcripts of different plasmids evince similar levels of mRNA, irrespectively of the plasmid’s translatability, cDNA dilution, and choice of PCR–primer pairs. This and the previous results suggest translation, particularly translation initiation, rather than transcription, as the decisive step in determining protein expression, which reverts the focus on the transcript sequence in the immediacy of the start codon.

In contrast to eukaryotic-based expression systems, the prokaryotic machinery is not capable of clearing the conformational elements of mRNA that may potentially hamper the correct assembly of the ribosome and hence of the initiation complex [21,22]. Although the specificity of the interaction between the ribosome and mRNA is mediated by the hybridization of the SD sequence and strengthened by the coupling of the first transfer-RNA (tRNA^Met^) to the start codon, the whole initiation complex extends over a much longer nucleotide segment. This segment or ribosome docking site (RDS) extends over 30 nucleotides downstream the SD sequence [10]. Since SD sequences are usually positioned five to 13 nucleotides before the start codon [21], the RDS extends into the coding sequence. Based on this fact, we changed our strategy of ameliorating constructs and opted for a quantitative approach. Inspired by the work of Na et al. [10], we developed a simple in silico translation–initiation potential model to quantify the likelihood of translation of a given mRNA sequence from a series of interaction energy parameters. The model defines the translation–initiation potential *δ* as
δ=exp(−(ΔESD+ΔEtRNA+ΔEopenRT))
where *R* is the Boltzmann constant, *T* is the temperature, Δ*E_SD_* is the hybridization energy between the SD and anti-SD sequences, Δ*E_tRNA_* is the hybridization energy of the start codon and its respective anti-codon (i.e., the tRNA^Met^), and Δ*E_open_* is the energy required to unfold the 30-nucleotide-long RDS. Here, Δ*E_SD_* and Δ*E_tRNA_* are constant, since neither the SD nor the start codon are altered. Consequently, variations in *δ* are exclusively determined by Δ*E_open_*. Applying the model to the plasmids under study enabled us to rationalize translation events, as translatable constructs consistently scored a higher *δ*, or a lower Δ*E_open_*, than the non-translatable ones. Figure 4a shows Δ*E_open_* as a function of the position of the SD sequence relative to the start codon, i. The graph displays a minimum about 11 nucleotides upstream from the start codon only in the case of translatable plasmids (Figure 4a). The deeper the minimum, the likelier the occurrence of protein expression.

In view of these results, Δ*E_open_* was used as a scoring function in a simulated-annealing algorithm to obtain a sequence that maximizes the accessibility of the RDS and preserves the 5′UTR and the native VDAC-coding sequence. Henceforth, we exploited the redundancy of the genetic code by introducing single-nucleotide, synonymous mutations in the VDAC-coding sequence. Applying the optimization algorithm on VDAC-II-A results in the construct VDAC-II-G, sporting seven synonymous mutations in the first nine TR codons. VDAC-II-G encodes the wild-type amino acid sequence of VDAC and displays a low value of Δ*E_open_* at the right location (Figure 4a). Figure 2c shows that VDAC-II-G experimentally enables protein expression in a comparable degree to those attained with enhancer-containing sequences.

Cell-free protein synthesis is governed by the biochemical conditions and the template DNA sequence. The *E. coli*-based system used in this study requires high concentrations of phage T7-RNA polymerase and a surplus of fast degradable amino acids, such as arginine, cysteine, tryptophan, glutamate, aspartate, and methionine [22,23]. Although necessary, these conditions are not as crucial in protein expression as the mRNA sequence, or rather, the mRNA conformational structure. Sequence elements in the proximity of the start codon, either upstream or downstream, are known to significantly affect translation efficiency [24,25,26], which implies not only finding the optimal location for the RBS [27] but also properly tailoring the whole RDS. Our findings are based on the design of several plasmids in which the sequences in the immediacy of the start codon have been altered to accommodate the RBS and the gene of a membrane protein at varying distances upstream and downstream the start codon, respectively. The results so far indicate that the best strategy for eliciting tag-free protein expression from constructs with off-the-shelf RBSs in prokaryotic cell-free expression systems entails the proper engineering of the TR proximal to the initiation codon.

Translation initiation in prokaryotes differs from that of eukaryotes in that it involves far fewer molecular factors and is significantly less complex. As pointed above, prokaryotes lack mRNA unfolding mechanisms that facilitate the formation of the translation initiation complex and hence are expected to rely on low-Δ*E_open_* transcripts to ensure the expression of their genes. Indeed, Δ*E_open_* at I ≈ −11 is significantly lower for transcripts of the *E. coli* genome than for those of the human genome. On the other hand, upregulation mechanisms for protein expression in prokaryotic cells are not present in cell-free systems and may be responsible for in-cell expression of recombinant VDAC from plasmids that do not otherwise elicit expression. Hence, the mRNA sequence is crucial in the cell-free context. Since the ribosome footprint on the mRNA sequence is larger than the RBS and extends well into the TR, a correspondingly long mRNA segment should be accessible for the ribosome to properly dock at, and initiate, translation. Hence, it makes sense to modify the mRNA sequence within the proximal TR so as to prevent the formation of hindering conformations and gain full access to the RDS. Thus, a low Δ*E_open_*(−11) can be viewed as a sine qua non criterion for cell-free protein expression with a prokaryotic machinery. According to Figure 4a, the efficiency in VDAC expression varies with the nature of the construct as follows: I-A > II-F ≅ II-G >> II-A. A trend that has been qualitatively confirmed by the experiments (Figure 1c and Figure 2c).

In this line, the role of translation enhancers in constructs with prokaryotic-like UTRs can be explained. Inserting human genes into pDEST14 vectors alone does not result in values of Δ*E_open_* low enough to allow expression (Figure 4b, red curve). Contrarily, the insertion of the 6xHIS-tag or the CAT enhancer nucleotide sequences significantly reduces Δ*E_open_* to similar or lower values than those of *E. coli* transcripts (Figure 4b, blue and purple curves). Thus, enhancers enable membrane protein expression inasmuch as they facilitate ribosome assembly through a less structured mRNA in the proximal TR.

Although both enhancers appear as valid options for constructs with poor translation efficiency, they may not be so in those cases where proteins with bare N-termini and native amino acid sequences are required [28]. Fortunately, the redundancy of the genetic code provides enough maneuverability to reduce Δ*E_open_* without altering the amino acid sequence, as shown in the case of VDAC. A potential working strategy that can be applied to any human membrane protein with prohibiting high Δ*E_open_* transcripts is reducing the magnitude to permissive prokaryotic values (Figure 4b, green curve). Hence, our results suggest that enhancing expression levels via sequence design optimization by synonymous mutations can be effectively employed in all those other cases where poor mRNA accessibility compromises the outset of translation, thus reducing protein expression.

## 4. Discussion

The current study demonstrates that the prokaryotic cell-free expression of human VDAC is determined by the mRNA sequence in the immediacy of the start codon and its impact on translation initiation. Providing the RBS site is optimal; i.e., 11 nucleotides upstream the ATG codon, the efficiency of protein expression can be enhanced by introducing synonymous mutations in the first nine codons of the TR. Computer calculations have provided a scoring function Δ*E_open_* that allows for a quantitative assessment of the translation initiation potential for the plasmids herein investigated, and an optimized, enhancer-free DNA sequence that allows for the cell-free expression of native VDAC is achieved. Thus, this computerized approach, publicly available as webservice XenoExpressO at www.rna.tbi.univie.ac.at (accessed on 6 July 2021), can predict the performance of plasmids in cell-free protein expression and provide an optimized sequence of translatable plasmids for VDAC and other human membrane proteins. One main advantage of this protein expression enhancement strategy is the easy implementation of silent mutations in expression vectors via primer design. Although challenges concerning the structure and function of membrane proteins still remain, our study presents a rational approach for an effective attempt at membrane protein synthesis efficiency.

## Figures and Tables

**Figure 1 membranes-11-00741-f001:**
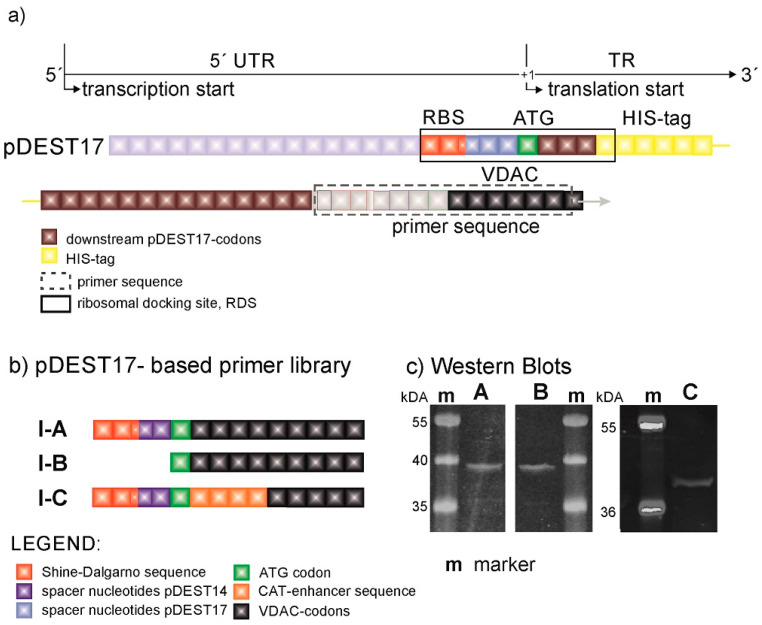
The nucleotide sequence pf PDEST17-based plasmids in the proximity of the start codon. (**a**) Primer library. (**b**) Expression levels (an SDS-page gel fluorescence scan) with a Safe Imager 2.0 TM Blue Light Transilluminator. The emission fluorescence at ≈470 nm of the fluorescent lysine accounted for the optical visualization of the protein bands. The respective expression levels were repeatedly achieved in several experiments with no significant difference in the expressed target protein to be observed (additional experiments are shown in the Appendix A). (**c**) nc: negative (no plasmid) control. To accommodate for the Gateway™ strategy, we inserted spacer nucleotides in the primer sequences, according to the manufacturer’s recommendation.

**Figure 2 membranes-11-00741-f002:**
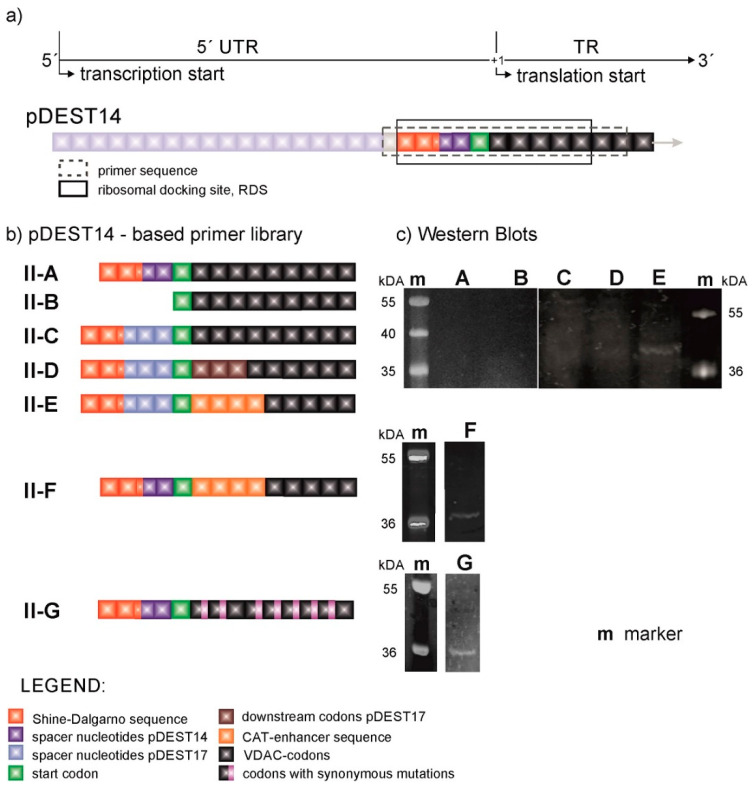
The nucleotide sequence of pDEST14-based plasmids in the proximity of the start codon. (**a**) Primer library. (**b**) Expression levels (SDS-page gel fluorescence scan) with a Safe Imager 2.0 TM Blue Light Transilluminator. The emission fluorescence at ≈470 nm of the fluorescent lysine accounted for the optical visualization of the protein bands. (**c**) nc: negative (no plasmid) control. Each construct was tested several times (at least three times), and the resulting protein amounts were tested as shown in the Appendix A by an Odyssey scan ™, demonstrating the reproducibility of the expression efficiencies.

**Figure 3 membranes-11-00741-f003:**
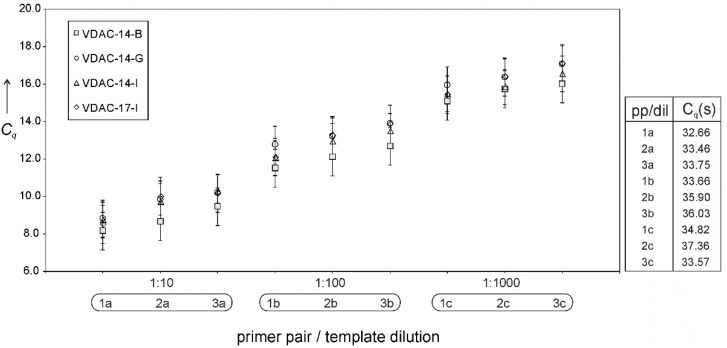
mRNA transcription levels (quantitation cycles, Cq) of different plasmids. Real-time amplification of three different dilutions of the as-obtained reverse-transcribed cDNA (a: 1:10; b: 1:100; c: 1:1000) with three different primer pairs (1, 2, and 3). Right: Cq values for the negative (no plasmid) control.

**Figure 4 membranes-11-00741-f004:**
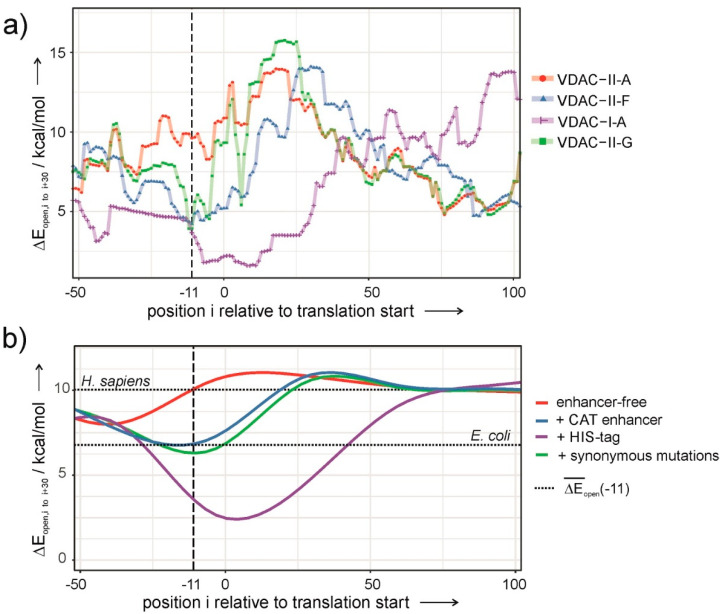
Δ*E_open_* (i) unfolding the 30-nucleotide-long RDS starting at location i with respect to the start codon (i = 0). (**a**) Translatable (VDAC-II-F,G, VDAC-I-A) and not-translatable constructs (VDAC-II-A) for VDAC expression. (**b**) Average Δ*E_open_* (i) for pDEST14 constructs of all human membrane proteins with and without expression enhancers and after coding sequence optimization with synonymous mutations. Dotted lines depict the values of Δ*E_open_* at i = −11 for the human and *E. coli* genomes.

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
