# Peer review of "Enhancing the Cell-Free Expression of Native Membrane Proteins by In Silico Optimization of the Coding Sequence—An Experimental Study of the Human Voltage-Dependent Anion Channel"

_membranes, 2021, doi:10.3390/membranes11100741_

Round 1

Reviewer 1 Report

The authors present the discussion of a relevant problem for in vitro protein synthesis and propose the alteration of the DNA sequence in the proximity of the start codon to improve the production of the protein. The authors do not show any evidence of the protein's functionality, which is not always directly proportional to the amount of protein expressed in vitro. For this reason, I believe this work is not too interesting for the scientist working in the cell-free protein synthesis field although the online tool for calculating the energy required to unfold the ribosome docking site could be useful for some researchers. The answers to the questions below must be eventually integrated into a revised version of the paper.  

What are the characteristics of pDEST17 for which you have was chosen this vector for in vitro protein synthesis?   

Why pDEST17 plasmid has those codons depicted in brown (upstream and downstream the HIS-tag sequence) in the illustration?

Why have you used primers in the library with and without the SD sequence? How are they supposed to hybridize to the plasmid considering that long brown sequence in pDEST17 before the gene of interest?  

In figure 1, to say that proteins express at the same level, you should measure the intensity of the bands produced using software for image analysis.  Upload figure (illustrations) of better quality.

Figure 1: why there are spacer nucleotides of pDEST14 in these constructs?

Figure 3 is of poor quality and cannot be read. Please upload an image of high quality suitable for publication. The current image is not suitable for publication.

Figure 4 are blurry please upload better figures.

Recall the supplementary materials in the main text to help the reader understanding better the logic you have used to draw the experiments and reach the conclusions.  

Reviewer 2 Report

The article Enhancing the cell-free expression of native membrane proteins by in-silico optimisation of the coding sequence - an experimental study of the human voltage-dependent anion channel is good one.

The concept of cell-free synthesis is herein employed to unravel the impact of the expression construct on gene transcription and translation,  without the complex regulatory mechanisms of cellular systems. But needs more study to prove the concept.

Is it possible to add some positive controls to the figure 1 and 2?

Is there any functional data?

How we know that the expressed proteins is working?

Round 2

Reviewer 1 Report

The authors must explain, in the manuscript, why they didn't perform the activity assay of the membrane protein expressed. I believe the expression of a protein without an activity assay is not completely interesting for scientists working on protein synthesis. Therefore, the authors must support their decision of not performing the activity assay in the work presented.  

Author Response

Respected Reviewer,

again, thank you very much for your time and effort to improve our manuscript. We have integrated into the introduction the connection to our previous work with out detailed electrochemical characterization to demonstrate functionality of the VDAC receptor, our example of choice.

We completely agree, that this link is necessary for the reader to value the potential of such in silico optimization for protein expression strategies (see line 75) .

with best regards,,

Eva Ehmoser 

Reviewer 2 Report

 minor correction in the grammar

Author Response

Dear Reviewer,

thank you very much again for improving the manuscript by your comments and suggestions.

We will prove-read the manuscript before final publication.

with best regards,

Eva Ehmoser